# Antisense transcription as a tool to tune gene expression

Jennifer AN Brophy & Christopher A Voigt[*]

## Abstract

A surprise that has emerged from transcriptomics is the prevalence of genomic antisense transcription, which occurs counter to gene orientation. While frequent, the roles of antisense transcription in regulation are poorly understood. We built a synthetic system in *Escherichia coli* to study how antisense transcription can change the expression of a gene and tune the response characteristics of a regulatory circuit. We developed a new genetic part that consists of a unidirectional terminator followed by a constitutive antisense promoter and demonstrate that this part represses gene expression proportionally to the antisense promoter strength. Chip-based oligo synthesis was applied to build a large library of 5,668 terminator–promoter combinations that was used to control the expression of three repressors (PhlF, SrpR, and TarA) in a simple genetic circuit (NOT gate). Using the library, we demonstrate that antisense promoters can be used to tune the threshold of a regulatory circuit without impacting other properties of its response function. Finally, we determined the relative contributions of antisense RNA and transcriptional interference to repressing gene expression and introduce a biophysical model to capture the impact of RNA polymerase collisions on gene repression. This work quantifies the role of antisense transcription in regulatory networks and introduces a new mode to control gene expression that has been previously overlooked in genetic engineering.

**Keywords** antisense transcription; design automation; genetic circuits; synthetic biology; systems biology
**Subject Categories** Synthetic Biology & Biotechnology; Quantitative Biology & Dynamical Systems; Transcription
**Mol Syst Biol. (2016) 12: 854**

## Introduction

Genetic engineers typically follow a simple scheme for controlling gene expression: a promoter and terminator flank the gene in the same orientation. Larger designs consisting of multiple genes are often organized similarly, where transcription is designed to proceed in one direction. This organization avoids potential interference between promoters that can arise due to RNA polymerase (RNAP) collisions (Shearwin *et al*, 2005), supercoiling (Pruss & Drlica, 1989; Chen & Wu, 2006), non-coding RNAs (Georg & Hess, 2011), and conflicts with the replication machinery (French, 1992; Vilette *et al*, 1995; Helmrich *et al*, 2013). Promoters that are oriented in the opposite direction of genes produce antisense transcription, and although this is generally regarded as a nuisance (Sharma *et al*, 2010; Singh *et al*, 2014), it can be useful for reducing leaky expression of toxic proteins (O'Connor & Timmis, 1987; Worrall & Connolly, 1990; Saida *et al*, 2006; Fozo *et al*, 2008) and generating genetic switches (Hongay *et al*, 2006; Chatterjee *et al*, 2011a,b). Here, we propose that promoters oriented opposite to a gene can be used reliably to tune gene expression and control the input threshold of genetic switches.

Increased use of transcriptomics has demonstrated that antisense transcription is surprisingly common across all organisms, including archaea (Wurtzel *et al*, 2010), prokaryotes (Selinger *et al*, 2000; Georg *et al*, 2009; Güell *et al*, 2009; Filiatrault *et al*, 2010; Wade & Grainger, 2014), and eukaryotes (Dujon, 1996; Yelin *et al*, 2003; He *et al*, 2008). For example, in *E. coli*, ~30% of all transcription start sites were found to be antisense and internal to, or just after, genes (Tutukina *et al*, 2007; Dornenburg *et al*, 2010; Thomason *et al*, 2015). Similarly in *H. pylori*, about half of the genes have at least one antisense promoter (Sharma *et al*, 2010). Although some of this antisense transcription is the result of inefficient termination by intrinsic and rho-dependent terminators (Peters *et al*, 2012), it is often driven by promoters with well-defined regulatory motifs, such as housekeeping sigma factor binding sites (Dornenburg *et al*, 2010; Raghavan *et al*, 2012; Wade & Grainger, 2014). Depending on the organism, antisense transcription can be constitutive or regulated under different environmental conditions (Beaume *et al*, 2010; Nicolas *et al*, 2012).

While prevalent, the role of most antisense transcription in regulation is unclear (Thomason & Storz, 2010; Sesto *et al*, 2013). Some have postulated that the majority of antisense transcription is non-functional and is background due to pervasive transcription (Raghavan *et al*, 2012). However, antisense transcription is known to be an important component of the genetic switches that control bacterial competence (Chatterjee *et al*, 2011b) and virulence (Mason *et al*, 2013), as well as *Saccharomyces cerevisiae*'s entry into meiosis (Hongay *et al*, 2006). It also occurs frequently for genes that require

Synthetic Biology Center, Department of Biological Engineering, Massachusetts Institute of Technology, Cambridge, MA, USA
*Corresponding author. Tel: +1 617 324 4851; E-mail: cavoigt@gmail.com

tight expression control under defined conditions, such as toxic or virulence proteins (Kawano *et al*, 2007; Fozo *et al*, 2008; Giangrossi *et al*, 2010; Lee & Groisman, 2010). One role of antisense transcription may be to impact the threshold of genetic switches (Liu & Kobayashi, 2007), defined as the amount of input signal required to reach half-maximal activity. It has been shown that the input threshold required for gene expression can be tuned by changing the translation efficiency of regulatory proteins, either by mutating ribosome binding sites or introducing small regulatory RNA, or by sequestering the proteins using dummy operators or protein–protein interactions (Buchler & Cross, 2009; Chen & Arkin, 2012; Lee & Maheshri, 2012; Rhodius *et al*, 2013). There is evidence that antisense promoters can similarly change regulatory circuits by controlling the expression of repressors, activators, σ factors, and anti-σ factors (Tutukina *et al*, 2010; Hirakawa *et al*, 2012).

There are two classes of mechanisms by which antisense promoters may regulate gene expression. The first involves the antisense RNA (asRNA) that is generated, which can regulate gene expression by binding to the mRNA to change its stability or translation, or act as a transcriptional regulator (Brantl & Wagner, 2002; Brantl, 2007). The second is transcriptional interference, where the sense and antisense promoters interact directly or via the RNAPs to cause the downregulation of a gene (Liu & Kobayashi, 2007). There are four mechanisms by which transcriptional interference can occur (Shearwin *et al*, 2005): (i) competition (promoters overlap and only one RNAP can bind at a time), (ii) sitting duck (an RNAP that is slow to elongate is dislodged), (iii) occlusion (one RNAP elongates over a promoter transiently blocking the other), and (iv) collision (two actively transcribing RNAPs collide) (Callen *et al*, 2004; Sneppen *et al*, 2005; Palmer *et al*, 2009). Of these, modeling suggests that when promoters are > 200 bp apart and oriented convergently, the dominant mechanism of interference is collision (Sneppen *et al*, 2005). Regulation by asRNA and transcriptional interference are not mutually exclusive. Examples have been described where regulation occurs due to only one mechanism (Liu & Kobayashi, 2007; André *et al*, 2008; Hirakawa *et al*, 2012) or they work in concert (Giangrossi *et al*, 2010; Chatterjee *et al*, 2011b).

In this work, we harness antisense transcription as a reliable "tuning knob" for the construction of genetic circuits. We introduce a new composite part to the 3'-end of the gene of interest that consists of a unidirectional terminator followed by a reverse constitutive promoter. We demonstrate that the antisense promoter represses gene expression in accordance with its strength and that the antisense transcription can cause a change in the threshold of inducible systems. A large library of promoter–terminator combinations was constructed via chip-based oligo synthesis (Tian *et al*, 2004) and screened using flow-seq to identify terminators and promoters that can be used to construct reliable antisense regulation (Sharon *et al*, 2012; Kosuri *et al*, 2013). This approach has been used previously to elucidate how translation rates affect mRNA stability (Kosuri *et al*, 2013) and codon bias influences RNA structure and translation (Goodman *et al*, 2013). Finally, we determined the relative contributions of antisense RNA and transcriptional interference to repressing gene expression and introduce a biophysical model to parameterize RNA polymerase collisions. This work contributes to a larger effort to expand on the classic concept of an "expression cassette" to include additional parts that utilize genetic

context to fine-tune expression levels (Lou *et al*, 2012; Mutalik *et al*, 2013; Brophy & Voigt, 2014).

# Results

## Repression correlates with the strength of the antisense promoter

A simple system was designed to quantify the impact of an antisense promoter on gene expression (Fig 1A). The isopropyl β-D-1-thiogalactopyranoside (IPTG)-inducible promoter $P_{tac}$ was used as

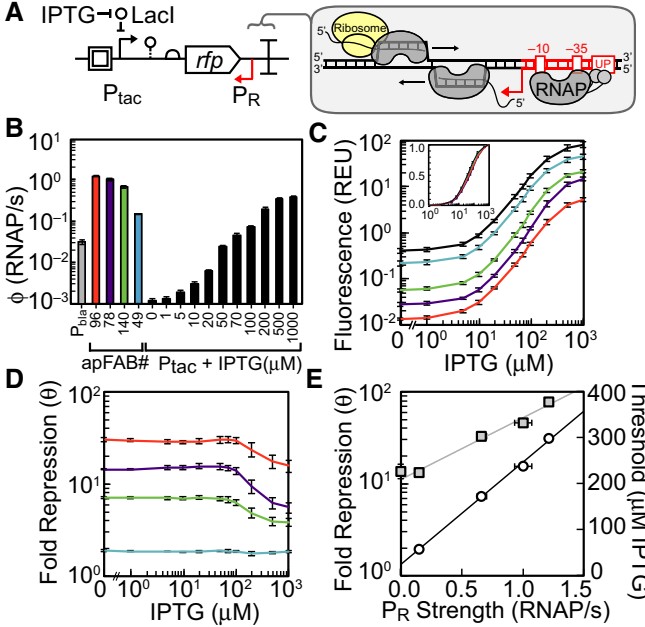

**Figure 1. Antisense transcription and its impact on gene expression.**

A   A schematic showing the antisense transcription reporter system. A constitutive promoter (red) at the 3'-end of *rfp* represses gene expression by firing polymerases at the forward promoter $P_{tac}$ (black).

B   Strengths of the constitutive promoters used as $P_R$ (colors) and the forward promoter ($P_{tac}$) at different inducer concentrations. The reference promoter ($P_{bla}$) used to calculate promoter strength in units of polymerase firings per second is shown (see Appendix for methods, promoter sequences, and plasmid maps).

C   Response functions for $P_{tac}$ with different antisense promoters located at the 3'-end of RFP: no promoter, black; antisense promoters of different strength, colors as in (B). The inset is the $\log_{10}$ transform of the same data normalized by min and max.

D   The fold repression (equation 1) is shown as a function of the induction of the forward promoter. The colors correspond to antisense promoters of different strength (B).

E   The maximum fold repression (white circles) or threshold (gray squares) is shown as a function of antisense promoter strength. The induction threshold K was calculated by fitting equation 2 to the data in (C). The lines are linear and exponential fits to the threshold ($R^2 = 0.9876$) and repression ($R^2 = 0.99737$) data, respectively.

Data information: In all panels, the data represent the mean of three experiments performed on different days, and the error bars are the standard deviation of these replicates.

Source data are available online for this figure.

the forward promoter to drive the expression of red fluorescent protein (RFP). Downstream of *rfp*, there is a constitutive antisense promoter $P_R$ whose RNAPs will be fired toward $P_{tac}$. The full cassette was placed on a plasmid containing the p15A origin (Appendix Fig S1). Four constitutive promoters of different strength (Kosuri *et al*, 2013; Mutalik *et al*, 2013) were selected to serve as the antisense promoters, and the impact on RFP expression was quantified. The strengths of $P_R$ and $P_{tac}$ were determined using a separate plasmid system and normalized by a reference standard to estimate promoter strengths as polymerase firing rates (Fig 1B) (Materials and Methods).

The addition of an antisense promoter changes the response curve for the IPTG induction of RFP in three ways (Fig 1C). First, expression is reduced over the entire range of inducer concentration. We defined a parameter $\theta$ that captures the magnitude of repression,

$$\theta = \frac{[RFP]_0}{[RFP]_+} \tag{1}$$

where the subscripts +/0 represent the presence/absence of an antisense promoter. Plotting $\theta$ versus inducer concentration shows that the impact of the antisense promoter is stronger when $P_{tac}$ is less active (Fig 1D). This biased repression is consistent with previous findings that weak promoters are more susceptible to repression via transcriptional interference and asRNAs than strong promoters (Callen *et al*, 2004; Shearwin *et al*, 2005; Sneppen *et al*, 2005). Second, the maximum repression increases with the strength of the antisense promoter (Fig 1E, exponential regression; $R^2 = 0.99737$). Notably, the strongest promoter tested (apFAB96) is unable to completely repress expression. This promoter is among the strongest from a large synthetic library (Kosuri *et al*, 2013) and of comparable strength to the *E. coli* rrn promoters (Liang *et al*, 1999). Finally, the threshold for induction increases as a function of the strength of antisense promoter (Fig 1E, linear regression; $R^2 = 0.9876$). Interestingly, the shape of the induction curve remains similar for the different antisense promoters (Fig 1C, inset). The cooperativity ($n = 1.7 \pm 0.1$) is also unaffected by different antisense promoters. Thresholds and Hill coefficients were calculated by fitting the response functions in Fig 1C to the Hill equation

$$y = y_{min} + (y_{max} - y_{min})\frac{x^n}{K^n + x^n} \tag{2}$$

where $x$ is the concentration of IPTG, $y$ is the activity of $P_{OUT}$, $n$ is the Hill coefficient, and $K$ is the threshold level of input where the output is half-maximal. *Escherichia coli* growth rates were also unaffected by the addition of antisense promoters (Appendix Fig S2).

### Multiplexed characterization of antisense promoters

Experiments were designed to quantify the impact of an antisense promoter on the function of a simple genetic circuit. We chose to characterize NOT gates, where an input promoter drives the expression of a repressor that turns off an output promoter (Yokobayashi *et al*, 2002). This creates a response function that is inverted compared to an inducible system alone. The NOT gate is a common logic motif that has been used to build more complex combinatorial logic and dynamic functions in living cells (Stanton *et al*, 2014). Our

design adds antisense promoters to the 3′-end of the repressor (Fig 2A), which reflects natural motifs where regulatory proteins are controlled by antisense promoters (Eiamphungporn & Helmann, 2009). The design also adds unidirectional terminators between the 3′-end of the repressor and the antisense promoter to demonstrate that antisense promoters can alter gene expression when added to the outside of complete expression cassettes.

Advances in chip-based DNA synthesis have made it possible to simultaneously synthesize 10,000s of unique ~200 bp oligos (Kosuri & Church, 2014). This length is appropriate to encode a terminator and antisense promoter. A library was constructed based on 52 terminators (Chen *et al*, 2013) and 109 constitutive promoters (Kosuri *et al*, 2013; Mutalik *et al*, 2013), paired combinatorially to produce 5,668 unique composite parts (Fig 2A). All of the promoters are synthetic and their strengths fall within a range of 0.0047 au to 21 au, with an average of 3.6 au (Kosuri *et al*, 2013). All of the terminators are naturally occurring sequences from the *E. coli* K12 genome and were selected to encompass a wide range of terminator strengths. The majority of these terminators are unidirectional and allow RNAPs fired from the antisense promoter to proceed while blocking those from the forward promoter (Chen *et al*, 2013). The composite parts were synthesized and cloned into three NOT gates made from TetR homologs (Materials and Methods) (Stanton *et al*, 2014). The NOT gate repressors (PhlF, SrpR, TarA) were selected to represent different response function shapes (Fig 2B).

The NOT gate libraries were screened using flow-seq, a technique where fluorescence-activated cell sorting (FACS) is used to sort the cells into bins, the contents of which are determined using next-generation sequencing (Materials and Methods) (Raveh-Sadka *et al*, 2012; Sharon *et al*, 2012). Here, we sorted the cells by NOT gate threshold, that is, the input promoter activity at which the output fluorescence is reduced to half-maximum. To do this, each library was grown with 100 μM IPTG and sorted by fluorescence into four log-spaced bins (Fig 2C). At 100 μM IPTG, all of the gates lacking antisense promoters are OFF (Fig 2B) and library members that are ON are likely to have increased induction thresholds. NOT gates without antisense promoters were grown with 0 and 100 μM IPTG to set upper and lower bounds for sorting, respectively. 6.5, 18.3 and 4.9% of the cells from the PhlF, SrpR, and TarA libraries have increased fluorescence relative to the gates without antisense promoters.

After sorting, cells were plated onto solid agar medium and eight colonies from each bin were randomly selected and their full response functions were measured (Fig 2D). As expected, the most fluorescent bins (bins 3 and 4) captured gates with the largest increase in threshold (Fig 2D, yellow and red). To quantify the effects of the antisense promoter/terminator parts, each response function was fit to equation 2.

Constructs from high fluorescence bins have increased thresholds $K$ relative to lower fluorescence bins, but the Hill coefficient $n$ does not show a consistent trend across bins (Fig EV1). After individual colonies were analyzed, we pooled the remaining cells from each bin (50,000–200,000 colonies) and measured their response functions in aggregate. Analysis of pooled constructs shows that the ON and OFF states of all library members are essentially constant and threshold differences between the bins are statistically significant (Fig EV2, one-way ANOVA).

After sorting, the bins were sequenced to identify the promoter/terminator combinations responsible for shifting gate

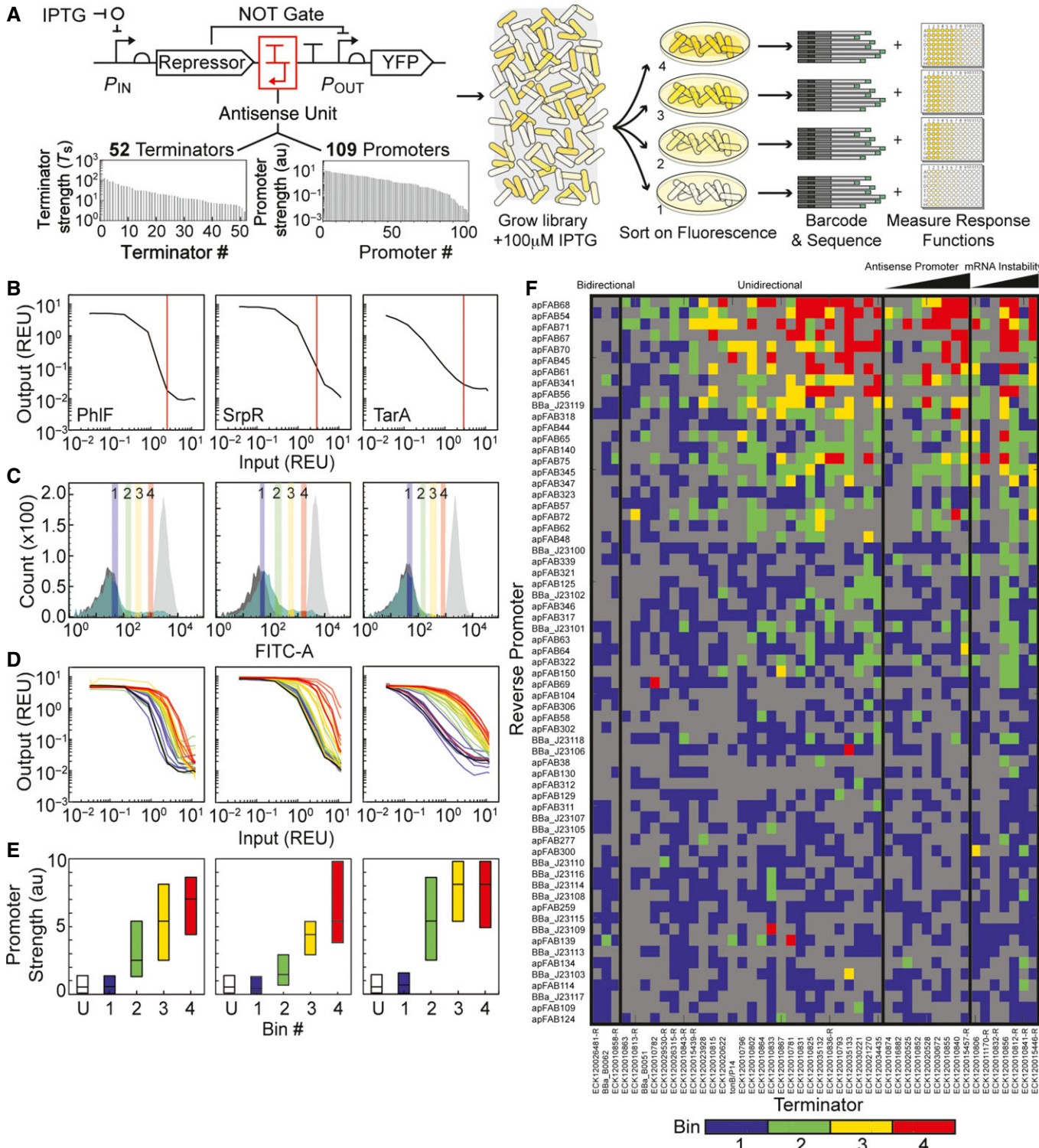

**Figure 2.**

thresholds. Briefly, plasmid DNA was isolated from bacteria in each bin, then the composite parts were amplified from the plasmids and barcoded for multiplexed sequencing. Paired-end reads were used to ensure complete sequencing of each promoter/terminator pair (Materials and Methods). For the following

analysis, we removed any sequencing reads that did not perfectly match the designed promoter/terminator pairs. The percent of perfect sequences from the pools was 32.38%, which is consistent with the error rate of chip-based oligo synthesis (Kosuri & Church, 2014).

◀

**Figure 2.  Construction of a library of terminator/antisense promoter pairs and characterization of their impact on regulatory circuit performance.**

A   Library construction and flow-seq screening used to measure the impact of terminator/antisense promoter pairs on NOT gate performance. All combinations of 52 unidirectional terminators and 109 promoters were constructed to create a library of 5,668 transcriptional interference constructs. The terminator and promoter strengths shown were measured previously (Chen *et al*, 2013; Kosuri *et al*, 2013; Mutalik *et al*, 2013). The library was synthesized as oligonucleotides, then cloned into genetic NOT gates at the 3′-end of the repressor gene (red box). Each library was transformed into *Escherichia coli*, grown with 100 μM IPTG, and sorted into bins of varying YFP fluorescence to find constructs with increased induction thresholds. Bacteria from each bin were plated on solid media, and individual colonies were selected to measure the full response function of sorted variants. Plates were scraped to isolate plasmids from bacteria in each bin, and plasmid DNA was barcoded and deep-sequenced (Materials and Methods).

B   Response functions of the starting NOT gates (no antisense promoters) built with TetR homologs: PhlF (left), SrpR (center), TarA (right). The response functions are measured using $P_{tac}$ activity as the input and YFP as the output. $P_{tac}$ activity was measured using a separate plasmid system and converted into REU with a reference standard (see Appendix for plasmid maps). Vertical lines (red) demarcate $P_{tac}$ activity with 100 μM IPTG, the inducer concentration used to sort libraries.

C   Fluorescence histograms of the unmodified NOT gates and libraries before sorting. Each unmodified NOT gate was grown with 0 μM (light gray) and 100 μM (dark gray) IPTG to set upper and lower bounds for sorting, respectively. Libraries (green) were sorted into four bins, shown as colored vertical bars and numbered by increasing fluorescence.

D   Response functions of twenty four randomly selected clones from the sorted libraries colored by the bin in which they were found (coloring is same as in C). The response functions of the NOT gates lacking antisense promoters are shown in black.

E   Deep sequencing revealed the identities of antisense promoters in each bin. Previously measured values, shown in (A), were used to calculate the median promoter strength of parts sorted into each bin. Box plots display the median, with hinges indicating the first and third quartiles. The unsorted library is marked "U".

F   Heat map shows the bin in which each terminator/promoter pair was most enriched. Promoters are rank ordered by their strength, with the strongest at the top. The terminators were grouped based on known or predicted terminator features and sorted by the predicted strength of a cryptic antisense promoter or impact on mRNA stability, if relevant (Materials and Methods). Unidirectional terminators were sorted based on similarity in their profiles across the promoter set. Terminator/promoter combinations that are not enriched in any of the bins are colored gray. Columns and rows > 90% gray were removed from the enrichment grid.

Source data are available online for this figure.

### The response threshold correlates with antisense promoter strength

Previously measured values of the promoter and terminator strengths were used to analyze the parts identified in each bin by deep sequencing (Fig 2A). In all three libraries, antisense promoter strength increases as a function of bin fluorescence (Appendix Fig S3, linear regressions; $R^2$ = 0.87421–0.96112). The high fluorescence bins (bins 3 and 4) contain constructs with greater median antisense promoter strength than lower fluorescence bins (bins 1 and 2) (Fig 2E). In contrast, there is no consistent trend in the forward or antisense terminator strengths across the sorted libraries (Appendix Figs S3 and S4).

The parts responsible for shifting gate thresholds were further explored by enrichment analysis. Enrichment identifies the parts that are selected for, or against, in each bin during sorting (Materials and Methods). Since high fluorescence bins (bins 3 and 4) have an increased threshold relative to lower fluorescence bins (bins 1 and 2) (Fig EV1), composite parts that are enriched in high fluorescence bins are more likely to generate large shifts in gate thresholds than those enriched in lower fluorescence bins. To visualize trends in the data, enrichment was used to assign each composite part to the one bin (1–4) that best reflects its ability to shift gate thresholds (Fig 2F). Most composite parts with strong antisense promoters are maximally enriched in bins 3 and 4 (Fig 2F, top). However, when strong promoters are paired with bidirectional terminators that have significant antisense termination efficiencies ($T_s$ antisense > 10), the composite parts are incapable of shifting circuit thresholds (Fig 2F, left). In contrast, when strong antisense promoters are paired with unidirectional terminators, the composite parts are sorted into high fluorescence bins. These terminators ($T_S$ antisense < 10) most likely facilitate greater shifts in gate thresholds than bidirectional terminators by allowing more RNAPs fired from the antisense promoter to proceed. Terminators that are predicted to destabilize mRNA or contain cryptic antisense promoters also facilitate large shifts in gate thresholds (Fig 2F, right). Changes in mRNA

stability can result in a large shift because the mRNA produces less protein before it is degraded (Chen *et al*, 2013); thus, more transcripts are required to produce the threshold amount of repressor protein. Similarly, terminators with cryptic antisense promoters increase the gate threshold by increasing the basal level of antisense transcription.

### Characterization of terminator/promoter pairs as "parts"

One of our goals is to use antisense transcription to reliably change the expression level of a gene or shift the threshold of a genetic circuit. Ideally, the impact of an antisense promoter on these functions would be predictable and a set of promoters of different strengths could be used to tune expression. When building multi-gene systems, it is desirable to use different terminators to control each gene in order to avoid homologous recombination (Sleight & Sauro, 2013). Therefore, we sought to identify a set of strong terminators that could be used in conjunction with a set of antisense promoters to reliably tune gene expression. Predictability would require that the promoters impart their effect independent of the terminator to which they are paired.

Some terminators may have mechanisms that impact the effectiveness of the antisense promoter. As such, we eliminated those with known features (cryptic promoters, bidirectional termination, hairpins that impact mRNA stability) from the set (Fig 2F). Then, terminators and promoters were systematically removed until there remained a core set of both in which the promoters produced a reliable response when combined with any of the terminators. This set, shown in Fig 3A, provides nine strong terminators that can be fused to different genes or operons and twenty antisense promoters that can be added to control their expression (Fig 3B). To confirm predictability, several terminator–promoter pairs were tested in a reporter construct with $P_{tac}$ and *yfp*. The repression produced by these pairs collapse onto a single curve, independent of the identity of the terminator used (Fig 3C, exponential regression; $R^2$ = 0.9275) (Appendix Table S2).

### Contribution of asRNA to repression

To determine the relative contributions of asRNA and transcriptional interference to repression generated by antisense promoters, we built a set of plasmids to express asRNA corresponding to the reporter gene (Appendix Fig S5). These plasmids each have one of the four antisense promoters (apFAB49, apFAB140, apFAB78, or apFAB96) driving expression of the reverse compliment of RFP followed by a strong bidirectional terminator (ECK120034435). The cassette is placed on a plasmid containing the ColE1 origin, which is maintained at a copy number approximately two times higher than p15A (Deininger, 1990). The asRNA plasmids were co-transformed into *E. coli* along with the original RFP reporter plasmid (pJBTI241, Appendix Fig S1).

Using these data, the fold repression due to antisense RNA $\theta_{asRNA}$ is calculated as in equation 1, where the subscripts +/0 now represent the presence or absence of *trans* encoded asRNA. $\theta_{asRNA}$ should be viewed as an upper bound on the contribution from asRNA to total fold repression $\theta$. This is because its expression in *trans* causes the asRNA to be longer (relative to the asRNAs generated at the 3′-end by RNAP collisions) and expressed at a higher level (due to the absence of RNAP collisions and the higher copy number plasmid). Fold repression generated by transcriptional interference $\theta_{TI}$ was then determined by dividing the total fold repression $\theta$ by $\theta_{asRNA}$. Thus, $\theta_{asRNA}$ and $\theta_{TI}$ reflect the relative contributions of asRNA and transcriptional interference to total repression generated by antisense transcription. Plotting $\theta$, $\theta_{asRNA}$, and $\theta_{TI}$ versus inducer concentration shows that asRNA and transcriptional interference generate approximately equivalent contributions to repression (Fig 4A, Appendix Fig S13). When the forward promoter is strongly induced, the predicted *cis* contribution declines, which is consistent with models of transcriptional interference (Sneppen *et al*, 2005) (Fig 4A, Appendix Fig S13). It is noteworthy that strong repression cannot be achieved through either transcriptional interference or asRNA alone. They each contribute to the total repression that can be achieved using antisense promoters. The *cis* regulation is likely most important for achieving maximal repression when the asRNA does not have specific regulatory qualities, such as RNA–RNA or RNA–DNA interaction hairpins (Lucks *et al*, 2011; Mutalik *et al*, 2012), Hfq binding motifs (Brennan & Link, 2007), or RNase processing sites (Lee & Groisman, 2010; Stazic *et al*, 2011).

### A quantitative model of transcriptional interference

A differential equation model was developed to explore collision interference and parameterize the repression that arises for different forward and antisense promoter firing rates ($\phi_F$ and $\phi_R$) and gene length $N$. The model and parameters are shown in Fig 4B and Appendix Table S3. Polymerases that originate from the forward promoter $P_F$ transcribe a gene at a constant velocity $v$ unless they collide with polymerases from the interfering promoter $P_R$ on the opposing strand of DNA. In the event of a collision, polymerases may dissociate from the DNA.

Collision interference can be captured by two differential equations that track the steady-state concentration of polymerases on the forward $C_F$ and reverse $C_R$ strands as a function of the distance from the start site $x$ (in bp):

$$\frac{dC_F}{dx} = -\varepsilon_F C_F C_R \tag{3}$$

$$\frac{dC_R}{dx} = \varepsilon_R C_F C_R \tag{4}$$

Here, the $\varepsilon_F$ and $\varepsilon_R$ are parameters that reflect the possibility that RNAPs fired from the forward and antisense promoters encounter collisions and dissociate from the DNA. Several factors may influence the frequency with which RNAPs encounter collisions and dissociate from the DNA. For example, transcriptional bursting could prevent RNAPs from encountering head-on collisions by increasing the time between transcription events initiated at the opposing promoter. *In vitro* experiments show that head-on RNAP collisions result in stalling and backtracking of the enzymes (Crampton *et al*, 2006), which leaves them vulnerable to clearance (Roberts

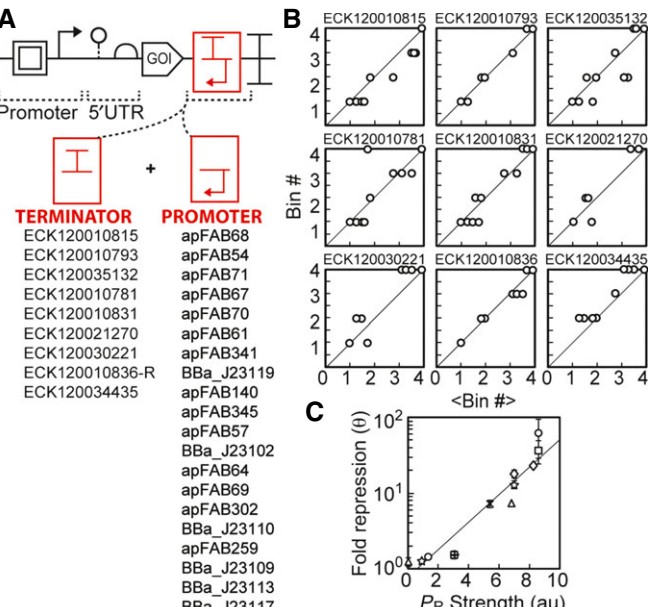

**Figure 3. Composability of unidirectional terminators and antisense promoters.**

A   The flow-seq data were used to identify a subset of promoters and terminators that could be combined to obtain a reliable reduction in gene expression.

B   Each graph shows a terminator (name at top), and each point is a promoter from the list in (A). The *x*-axis (<Bin#>) is the average for the promoter across the complete terminator set, and the *y*-axis (Bin#) is the bin for the specific terminator. The Bin# is calculated as described in Fig 2F.

C   Repression was explicitly measured for a subset of terminator/promoter pairs selected from (A). The pairs were cloned into the plasmid from (A) with P$_{tac}$ and *yfp*, and fold repression (equation 1) was measured as a function of the forward promoter activity (see Appendix Table S1 for terminator/promoter combinations tested). Maximum fold repression is plotted against the previously measured promoter activities (Kosuri *et al*, 2013); $R^2$ = 0.9275. Composite parts are marked by terminator (ECK120035132, circle; ECK120034435, triangle; ECK120021270, diamond; ECK120010793, star; ECK120030221, x; ECK120010815, +). Data represent the mean of three experiments performed on different days, and the error bars are the standard deviation of these measurements.

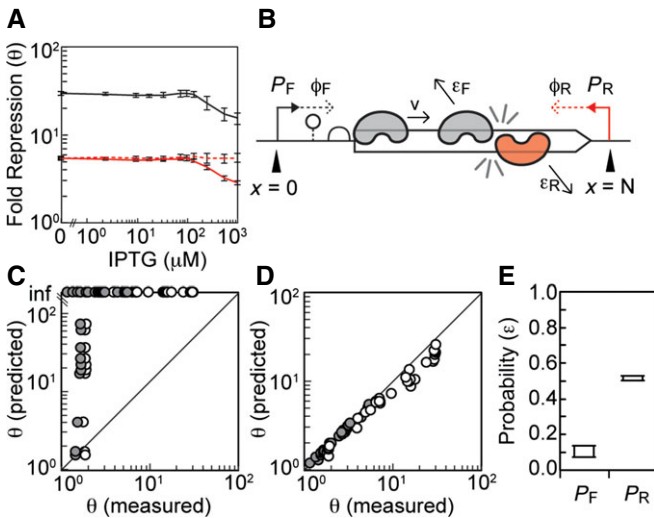

**Figure 4.  Mechanisms of repression by antisense transcription.**

A   The total fold repression θ generated by the apFAB96 antisense promoter is shown as a function of forward promoter activity using the characterization system in Fig 1A (black line). This is compared to the repression observed when the same promoter is used to drive the transcription of asRNA in *trans* from a separate plasmid (dashed red line) (Appendix Fig S5). The repression due to transcriptional interference θ_TI (solid red line) is inferred from the total and *trans* asRNA repression data (see text for details).

B   Schematic of the transcriptional interference model. A forward P_F and antisense P_R promoter are located on either side of a gene, N bases apart. The promoters fire at rates φ_F and φ_R RNAP/second. Polymerases transcribe at a constant velocity υ unless they collide with polymerases fired from opposing promoter. Polymerases collide and dissociate from the DNA with a probability ε.

C   Comparison of model predictions to experimental data when ε_F = ε_R = 1. Each data point shows experimentally measured repression (θ - white, θ_TI - gray) for each P_F/P_R pair plotted against model predictions generated with the same promoter combinations. Repressions predicted by the model were calculated using equation 5.

D   Comparison of model predictions to experimental data for the highest scoring ε_F and ε_R. The optimal ε_F and ε_R values simulate repression closest to the experimentally measured value (Materials and Methods). Data points show experimentally measured repression (θ – white, θ_TI – gray) for each P_F/P_R pair plotted against model results generated with the optimal ε_F and ε_R values for each promoter combination.

E   The range of optimal ε values that result from fitting the model to θ (which assumes that transcriptional interference is the sole mechanism of repression) or θ_TI (which is the minimum amount of repression attributable to transcriptional interference). Box plot extends from the median optimal ε_F and ε_R when the model is fit to θ_TI (⟨ε_F⟩ = 0.07, ⟨ε_R⟩ = 0.52) to median optimal ε_F and ε_R when the model is fit to θ (⟨ε_F⟩ = 0.14, ⟨ε_R⟩ = 0.47).

Data information: In all panels, the data represent the mean of three experiments performed on different days, and the error bars are the standard deviation of these replicates. Fold repression θ is the magnitude of gene expression produced by forward promoter in the absence of an antisense promoter divided by the amount of gene expression produced in the presence of an antisense promoter.

Source data are available online for this figure.

& Park, 2004; Nudler, 2012). However, co-translating ribosomes (Proshkin *et al*, 2010) and actively transcribing RNAPs (Epshtein & Nudler, 2003; Epshtein *et al*, 2003) have been shown to rescue stalled/backtracked complexes by realigning the 3′-termini of the RNA transcript with the enzyme's active site. In addition, there are active mechanisms that favor the termination and release of RNAPs

transcribing non-coding RNA (Kaplan & O'Donnell, 2003). Therefore, the model was built to accommodate the differences in dissociation for RNAPs fired from forward and antisense promoters of different strengths.

Boundary conditions are defined by the rates that polymerases are fired, that is, begin elongating, at the forward $C_F(x = 0) = φ_F/v$ and reverse $C_R(x = N) = φ_R/v$ promoters. The polymerase velocity $v = 40$ bp/s is held constant (Vogel & Jensen, 1994; Sneppen *et al*, 2005). The equations are numerically solved for each $φ_F$ and $φ_R$ combination (Materials and Methods). We chose to model $P_F$ as $P_{tac}$ with ten different IPTG concentrations and $P_R$ as four different constitutive promoters: apFAB49, apFAB140, apFAB78, and apFAB96 (Fig 1B). Simulated repression can be calculated as the ratio of full-length ($x = N$) transcripts produced from $P_F$ with and without an antisense promoter

$$θ = \frac{C_F|ε_F = 0}{C_F|ε_F > 0}, \tag{5}$$

providing a prediction that can be compared with measurements (equation 1). For each combination of forward and antisense promoters, $θ$ is calculated and compared with that derived from experiments.

Repression due to asRNA is not included in the model. Rather, our approach was to fit the model predictions to the two bounds that we measured. First, results were fit to total repression $θ$, which assumes that there is no contribution from asRNA to the observed repression. Next results were fit to $θ_{TI}$, which represents the minimum amount of repression attributable to transcriptional interference. These bounds were then used to fit the underlying biophysical parameters and provide a range of values that reflect the possible contribution of transcriptional interference to gene repression.

We first simulated collision interference with $ε_F = ε_R = 1$, since previous models of transcriptional interference assume that actively transcribing polymerases never survive head-on collisions (Sneppen *et al*, 2005; Chatterjee *et al*, 2011a,b) (Fig 4C). However, assuming $ε_F = ε_R = 1$ predicts too much repression and results in a poor fit to our experiments. To optimize $ε_F$ and $ε_R$, the model was solved where these parameters are varied in the range [0,1] in increments of 0.01 (Appendix Fig S6). This was repeated for each value of $φ_F$ and $φ_R$ in our data set, and the values of $ε_F$ and $ε_R$ that generate repression closest to the experimentally measured $θ_{TI}$ were determined. This yielded a set of 80 optimal values of $ε_F$ and $ε_R$, corresponding to all of combinations of forward and antisense promoter activities (40 pairings) fit to either $θ$ or $θ_{TI}$. The optimal values of $ε_F$ and $ε_R$ produced behavior that closely matches the experimental data (Fig 4D, linear regression; $m = 1$, $R^2 = 0.84082$). Values of $ε < 1$ are interpreted as cases where polymerases either avoid or survive collision and continue transcribing. Importantly, the model does not assume that one RNAP must dissociate in order for the RNAP on the opposing strand of DNA to survive collision. This ability to bypass is supported by *in vitro* experiments done with viral (Ma & McAllister, 2009) and yeast (Hobson *et al*, 2012) RNA polymerases.

The optimum values of $ε_F$ and $ε_R$ are surprisingly constant across the dataset and are independent of the identity of the antisense promoter or the firing rate of the forward promoter (Appendix Fig S7). $ε$ values fit using $θ$ represent the probabilities that polymerases collide and dissociate when transcriptional interference is assumed

to be the sole mechanism for gene repression. In contrast, $\varepsilon$ values fit using $\theta_{TI}$ reflect the smallest possible contribution of transcriptional interference to gene repression. Thus, we find a range of $\varepsilon$ values that reflect the potential contributions of collision interference to gene repression. We find that the probability that a polymerase dissociates due to a competing polymerase is significantly larger for RNAPs fired from the antisense promoter ($<\varepsilon_R> = 43$–$54\%$) as compared to those from the forward promoter ($<\varepsilon_F> = 7$–$15\%$) (Fig 4E). In addition, the model predicts that fold repression due to transcriptional interference increases exponentially as a function of the antisense promoter strength and distance between the two promoters (Fig EV3A and B). To test repression as a function of distance between the forward and antisense promoters, we modified our antisense reporter system and inserted *yfp* between the 3′-end of *rfp* and the antisense promoter (Fig EV3D). This increases the distance between the two promoters from 850 to 1,500 bp. Measuring repression of RFP and YFP using this system shows that repression increases as the distance between the two promoters grows.

## Discussion

This work demonstrates that antisense promoters can be reliably used to tune gene expression. The degree of repression is proportional to the strength of the antisense promoter over a > 30-fold range. This builds on the modern revisiting of the classical "expression cassette" to incorporate additional non-canonical parts to tune expression, insulate against context, and provide for rapid debugging via omics techniques (Mutalik *et al*, 2013; Nielsen *et al*, 2013). In this paradigm, there are alternative means to control the expression level of a gene, such as changing the forward promoter (Mutalik *et al*, 2013) or RBS (Salis *et al*, 2009), or adding small RNA (Lucks *et al*, 2011; Mutalik *et al*, 2012; Na *et al*, 2013) or 3′-hairpins to alter mRNA stability (Carrier & Keasling, 1997). While some of these approaches can achieve greater ranges of expression control, antisense promoters have unique features that are advantageous for some applications. Notably, they offer a means of control external to the expression cassette. This is particularly valuable when the forward control elements (promoter and RBS) have been engineered to integrate additional regulatory information (Cox *et al*, 2007; Carothers *et al*, 2011; Stanton *et al*, 2014; Stevens & Carothers, 2015). In these cases, it is not simple to adjust the overall expression level without interfering with how the signals are integrated. Exploiting antisense transcription allows for control without changing the inputs to the system or the sequence of the forward transcript. In addition, our flow-seq data demonstrate that the impact of the antisense promoter is largely context independent. From this, we derived a set of unidirectional terminators that can be combined with the antisense promoter in a modular manner. Thus, implementing this control is simple and modular and can be done with existing promoter libraries.

The performance of antisense transcription is derived from its unique synergy between its impact on transcription from the forward promoter and post-transcriptional impact on protein expression. For our system, we find that antisense RNA and collision between actively transcribing RNA polymerases contribute roughly equally to repress gene expression. This synergy is important because transcriptional interference implements its control at the transcriptional level and would not be able to repress mRNAs made by RNAPs that avoid collision. Antisense RNAs prohibit escaped mRNA transcripts from being translated. For asRNAs with weak affinity for the target mRNA, cohesion between the two mechanisms may facilitate greater repression of target genes than the asRNA alone.

The model predicts some mechanistic details about collision interference. Most strikingly, polymerases transcribing translated mRNA survive or avoid ~85% of their head-on collisions, which may explain the inability to completely abolish gene expression with transcriptional interference alone, despite the use of very strong interfering promoters. Polymerase survival rates are not as high if the polymerase is fired from the antisense promoter (~50% survival). This imbalance may be due to the differences in the kinetic properties of the two promoters, for example, burstiness and differential dissociation of the RNAPs. Single-molecule experiments that measure polymerase survival rates directly could be done to differentiate between these two mechanisms. Additional experiments can also refine our understanding of antisense transcription by parameterizing additional mechanisms. The model presented here is limited to collision interference as the mechanism of repression in *cis*; however, several additional factors, such as r-loop formation (Gowrishankar *et al*, 2013), changes in DNA topology (Chen & Wu, 2006), differences in local asRNA concentration (Llopis *et al*, 2010), and occlusion interference (Palmer *et al*, 2009), could also be considered. Direct measurement of asRNA activity in *cis* and *trans*, as well as the measurement of RNA duplex formation and degradation rates, r-loop formation, and other modes of transcriptional interference, would facilitate the construction of a more detailed mechanistic model of repression mediated by antisense transcription.

Considering natural genomes, antisense promoters could be a simple evolutionary mechanism to tune gene expression. Since housekeeping sigma factors—such as $\sigma_{70}$—have relatively information poor binding sites, promoters are expected to frequently arise spontaneously during evolution (Raghavan *et al*, 2012). Thus, constitutive antisense promoters could provide a simple mechanism to reduce gene expression that can be rapidly discovered during an evolutionary search. This is consistent with the lack of conservation of antisense promoters between species (Nicolas *et al*, 2012; Raghavan *et al*, 2012), which appear and disappear quickly in evolutionary time to fine-tune expression. Antisense transcription would be an easier solution to find than making mutations to the sense promoter, which can be significantly constrained by needs of regulatory signal integration (Price *et al*, 2005). Thus, the total expression of a gene could be tuned without disturbing the integration of signals. The rate of evolution around any individual promoters may not be high (Raghavan *et al*, 2012) because there are many similar solutions that can be found.

Here, we show how antisense transcription can be integrated into a simple NOT gate, which has been a common motif in building larger synthetic genetic circuits. Antisense transcription provides a mechanism where the switching threshold can be tuned without impacting other characteristics of the gate, such as the cooperativity. More complex circuits could be built by exploiting antisense transcription in both prokaryotic and eukaryotic systems where antisense regulation is known to occur (Hongay *et al*, 2006; Chatterjee *et al*, 2011b; Mason *et al*, 2013) and additional tuning knobs can

help improve the performance of synthetic systems (Murphy *et al*, 2010; Nevozhay *et al*, 2013; Brophy & Voigt, 2014). The simple constitutive promoters we employ here could be exchanged for dynamic promoters that respond to inducers or cellular/environmental conditions or implement negative feedback. This occurs in natural regulatory networks; for example, many of the antisense promoters in *B. subtilis* are regulated by alternative sigma factors that respond to different environmental conditions (Nicolas *et al*, 2012). This gets more complex as the sigma factors themselves are regulated by antisense transcription (Eiamphungporn & Helmann, 2009). Even more interesting architectures have been observed in nature; for example, there are many that involve overlapping 5′- and 3′-UTRs. The overlap can include entire genes; for example, divergent operons have been observed where the promoter for each occurs one gene into the other (Chatterjee *et al*, 2011b; Lasa *et al*, 2011). These motifs would enable mutually exclusive switch-like changes between the sets of genes that are expressed (Sesto *et al*, 2013). Collectively, this points to antisense transcription as something that should be routinely incorporated into engineered systems, as opposed to being avoided.

# Materials and Methods

### Strains and media

*Escherichia coli* strains NEB10β (Δ*(ara-leu)7697 araD139 fhuA ΔlacX74 galK16 galE15 e14- φ80dlacZΔM15 recA1 relA1 endA1 nupG rpsL* (Str^R) *rph spoT1 Δ(mrr-hsdRMS-mcrBC)*) and DH10B (*F⁻ Δ(ara-leu)7697 araD139 ΔlacX74 galE15 φ80dlacZΔM15 recA1 endA1 nupG rpsL mcrA Δ(mrr-hsdRMS-mcrBC) λ⁻*) were used for all experiments. Cells were grown in either LB Miller broth (Becton Dickinson 244630) or M9 minimal medium supplemented with glucose (6.8 g/l $Na_2HPO_4$, 3 g/l $KH_2PO_4$, 0.5 g/l NaCl, 1 g/l $NH_4Cl$; Sigma M6030), 2 mM $MgSO_4$ (Affymetrix 18651), 100 μM $CaCl_2$ (Sigma C1016), 0.4% glucose (Fisher scientific M10046), 0.2% casamino acids (Becton Dickinson 223050), 340 mg/ml thiamine (vitamin B1) (Alfa Aesar A19560). Carbenicillin (100 μg/ml) (Gold Bio C-103), kanamycin (50 μg/ml) (Gold Bio K-120), and/or chloramphenicol (35 μg/ml) (USB Corporation 23660) were added to growth media to maintain plasmids when appropriate. Isopropyl β-D-1-thiogalactopyranoside (IPTG) (Gold Bio I2481C) was used as the inducer for all constructs.

### Measurement of response functions

*Escherichia coli* strains were grown for 16 h in LB media containing antibiotics in 96-deep well blocks (USA Scientific 1896–2000) at 37°C and 250 rpm in an INFORS-HT Multitron Pro. After 16 h, the cultures were diluted 1:200 into M9 medium with antibiotics and grown for 3 h with the same shaking and temperature settings as the overnight growth. Next, the cultures were diluted 1:700 into fresh M9 medium with antibiotics and different concentrations of isopropyl β-D-1-thiogalactopyranoside (IPTG). These cultures were grown for 6 h and then diluted 1:5 into phosphate-buffered saline (PBS) containing 2 mg/ml kanamycin or 35 μg/ml chloramphenicol to arrest protein production, and fluorescence was measured using a flow cytometer.

### Cytometry measurement and data analysis

Cells were analyzed by flow cytometry using a BD Biosciences Fortessa flow cytometer with blue (488-nm) and red (640-nm) lasers. An injection volume of 10 μl and flow rate of 0.5 μl/s were used. Cytometry data were analyzed using FlowJo (TreeStar Inc., Ashland, OR), and populations were gated on forward and side scatter heights. The gated populations consisted of at least 30,000 cells. The median fluorescence of the gated populations was used calculated using FlowJo and used for all reporting. Autofluorescence of white cells (NEB10β without plasmids) was subtracted from all fluorescence measurements.

### Promoter strength calculations

Promoter firing rates (RNAP/second) were estimated using NEB10β cells harboring one of the following plasmids: pJBTI26, pJBTI264, pJBTI265, pJBTI266, pJBTI267, pJBTI136 (Appendix Fig S10). Fluorescence of each strain was measured as described above. Fluorescence produced by the strain harboring plasmid pJBTI136 (<YFP> = 528 au) was used to define a promoter firing rate of 0.031 RNAP/second, which has been reported for promoter $P_{bla}$ (Liang *et al*, 1999). Fluorescence of strains carrying the other plasmids was divided by fluorescence produced by the strain harboring pJBTI136 and multiplied by 0.031 RNAP/second to obtain promoter firing rates. The hammerhead ribozyme insulator RiboJ (Lou *et al*, 2012) was used to standardize the 5′-UTR of YFP mRNA so that changes in fluorescence could be attributed solely to differences in polymerase firing. To convert promoter firing rates (RNAP/second) back to arbitrary units reported by the cytometer, multiply the firing rates by 17,032.

Relative expression units (REUs) were calculated using DH10B cells harboring one of the following plasmids: 0RFP2, pAN1717 (Appendix Fig S11A). Strains harboring 0RFP2 and pAN1717 were grown and measured in parallel with experimental strains. To convert raw RFP fluorescence measurements into REU, RFP produced by experimental strains was divided by red fluorescence produced by the strain harboring 0RFP2. To convert our reported RFP measurements (REU) back to arbitrary units, multiply the REU value by 2,295. To convert raw YFP fluorescence measurements into REU, YFP produced by experimental strains was divided by YFP produced by pAN1717. To convert our reported YFP measurements (REU) back to arbitrary units, multiply the REU value by 550. When measuring NOT gate response functions (Fig 2B), input promoter ($P_{tac}$) activity was measured using plasmid pJBTI26 (Appendix Fig S11B) and converted into REUs as described here. Cellular autofluorescence was subtracted before conversion to RNAP/second or REU by measuring the fluorescence of unmodified NEB10b cells.

Promoter strength measurements reported throughout the paper as au are from the RNA-seq experiments of Kosuri *et al* (2013). In these experiments, promoter strengths are calculated using RNA-seq read depth of mRNA produced by each promoter driving the expression of green fluorescent protein.

### Classification of terminators

Terminators that encode cryptic antisense promoters or destabilize mRNA when placed at the 3′-end were identified by analyzing data

from Chen *et al*'s study of *E. coli* intrinsic terminators. In this study, termination strength was measured by observing the changes in GFP and RFP expression that occur when a terminator is placed between two fluorescent proteins (5′ GFP and 3′ RFP). Strong terminators resulted in a large drop in RFP fluorescence relative to a control plasmid with no terminator (pGR). Chen *et al* measured several terminators in both the forward and reverse orientation, which allowed us to identify unidirectional terminators for this study. We classified terminators as unidirectional if they have termination strength < 10 in the reverse orientation and > 10 in the forward direction.

Average levels of GFP and RFP fluorescence produced by plasmids carrying Chen *et al*'s library of terminators were used to identify terminators that encode cryptic antisense promoters or destabilize mRNA. Terminators that decreased GFP expression relative to the average were assumed to destabilize mRNA. Similarly, terminators that increased RFP expression when measured in the reverse direction were assumed to encode cryptic antisense promoters. We classified terminators that decreased GFP expression more than one standard deviation below the mean as destabilizing mRNA and terminators that increased RFP expression more than one standard deviation as encoding cryptic antisense promoters.

## Library design and construction

The terminator/antisense promoter library was built as described previously (Kosuri *et al*, 2013). The library was constructed based on 52 terminators (Chen *et al*, 2013) and 109 constitutive promoters (Kosuri *et al*, 2013; Mutalik *et al*, 2013), paired combinatorially to produce 5,668 unique composite parts. We used 90 promoters from an existing library (Mutalik *et al*, 2013) and 19 from the Anderson promoter library on the BioBricks Registry. The terminators are naturally occurring sequences from the *E. coli* K12 genome that were previously characterized by Chen *et al* (2013) and selected to encompass a wide range of terminator strengths. The composite parts were checked for restriction sites (NotI and SbfI) and none were found. To generate the final library, all sequences were flanked by restriction enzyme sites (NotI and SbfI) and PCR primer binding sites: (i) ATATAGATGCCGTCCTAGCG and (ii) AAGTATCT TTCCTGTGCCCA.

The oligonucleotide library was constructed by CustomArray, Inc., using their CMOS semiconductor technology. The library was delivered as a 1 fM oligonucleotide pool and amplified using specific PCR primers: oj1299 and oj1300 (Appendix Table S4). The PCR products were then digested with NotI (New England Biolabs R3189) and SbfI (New England Biolabs R3642) restriction enzymes and cleaned with DNA Clean & Concentrator columns (Zymo Research C1003). Plasmid backbones encoding repressor protein-based NOT gates (PhlF, SrpR, TarA; maps in Appendix Fig S9) were amplified by PCR with primers to add NotI and SbfI restriction sites to the 3′-end of the repressor gene. Plasmid backbones were then digested with the same restriction enzymes and cleaned using DNA Clean & Concentrator columns. After digestion, the library inserts and plasmid backbones were ligated using T4 DNA ligase (New England Biolabs M0208) and cloned into *E. coli* NEB10β-electrocompetent cells (New England Biolabs C3020K), resulting in three libraries (PhlF, SrpR, and TarA) of ~160,000 clones each and > 20-fold coverage of the designed sequence space. Each library was

scraped from solid media plates and frozen at −80°C in 200 μl aliquots with 15% glycerol for subsequent analysis.

## Library growth and fluorescence-activated cell sorting (FACS)

To grow libraries for flow cytometry analysis or cell sorting, one aliquot of each library was thawed and 10 μl of the sample was added to 3 ml of LB media supplemented with carbenicillin in 15-ml culture tubes (Fischer Scientific 352059). Once thawed, the remaining library aliquot was discarded to avoid cell death from repeated freeze–thaw cycles. The inoculated libraries were grown for 12 h at 30°C and 250 rpm in a New Brunswick Scientific Innova 44. NEB10β control strains containing unmodified NOT gate plasmids (Appendix Fig S9) were inoculated from single colonies into 3 ml of LB supplemented with carbenicillin and also grown at 30°C and 250 rpm After 12 h, both library and control strain cultures were diluted 1:200 into 25 ml of M9 medium with carbenicillin in 250-ml Erlenmeyer flasks (Corning 4,450–250) and grown at 37°C and 250 rpm for 3 h. Next, the cultures were diluted to 0.001 $OD_{600}$ in 25 ml of M9 medium with carbenicillin and either 0 or 100 μM IPTG. These cultures were grown for 6 h to obtain exponential phase growth. At the end of 6 h, cultures were diluted to $OD_{600}$ ~0.05 into PBS containing 35 μg/ml chloramphenicol to arrest cell growth and protein production until sorting. Aliquots of each library were also frozen at −80°C with 15% glycerol (VWR BDH1172) to serve as "unsorted" controls.

Cell sorting was done on a BD Biosciences FACSAria II with a blue (488-nm) laser. Each NOT gate library was sorted into four non-adjacent log-spaced bins based on YFP fluorescence. Control strains grown with 0 and 100 μM IPTG defined the upper and lower boundaries for bin placement, respectively. One million cells were sorted into the lowest fluorescence bin (Bin 1; Fig 2C, blue), which captured 9.8–13.9% of each library; 50,000–200,000 cells were sorted into all other bins, which captured 0.2–4.4% of the cells in each library. After sorting, cells were plated on solid media to minimize the effect of growth rate differences on library representation. Each bin was then scraped from the solid media plates and frozen at −80°C in 200 μl aliquots with 15% glycerol (VWR BDH1172) for subsequent analysis.

## Sorted library sequencing

Plasmids were isolated from cells in each bin using a miniprep kit (Qiagen 1018398) by thawing one aliquot of each frozen sorted bin and using the entire sample as input to the kit. For deep sequencing, 30 ng of each miniprepped sample was amplified for thirty cycles of PCR with Phusion High-Fidelity Polymerase Master Mix (New England Biolabs M0531). This PCR step added barcodes to each sample using primers oj1302, 1334 – oj1348 (Appendix Table S4). Amplification of samples was verified with gel electrophoresis and quantified using a NanoDrop spectrophotometer (ND-1000). Unsorted control samples were identically processed and sequenced. 13.1 M constructs were sequenced in a single MiSeq 150 paired-end lane with the sequencing primers oj1301, oj1303, and oj1356 (Appendix Table S4). To correct for the fact that fewer cells are sorted into the later bins (BIN2-4), the samples were mixed such that the "unsorted" and "BIN1" samples were present in equimolar ratios and made up 90% of the final sequenced mixture. The

"BIN2", "BIN3", and "BIN4" samples, which were also mixed in equimolar ratios, constituted the last 10% of the final sequenced mixture. This resulted in 1.7–2.2 million sequencing reads from the each of the "unsorted" and "BIN1" samples and 100,000–180,000 sequencing reads from each of the "BIN2", "BIN3", and "BIN4" samples (Appendix Table S1).

### Deep sequencing analysis

Custom software ("IlluminaSeqAnalysis.m", Code EV1) was written to combine paired-end reads and identify composite parts with perfect sequence identity to designed constructs. Each set of paired 150-bp reads was aligned and merged into a contig based on over-lapping sequence. NotI and SbfI restriction enzyme sites were identified, and all sequences (including adapter and constant primer sequence) outside the restriction sites were trimmed from both ends of the contig. Reads that did not pair or did not have both restriction sites were discarded since all composite parts were under 200 bp, and thus, paired reads should have overlapping sequence and yield contigs with both restriction sites. Of the 13.1 M constructs sequenced, 11.2 M (85.30%) yielded paired reads with overlapping sequence and both restriction sites. Once paired, all remaining contigs with mismatches (insertions, deletions, or substitutions) to designed constructs were discarded. Of the 11.2 M contigs, 3.6 M (32.28%) are perfect matches to the designed library. This is consistent with the error rate of chip-based oligo synthesis (Kosuri & Church, 2014).

Analysis of the perfect sequences shows that 70.0–77.5% of the composite parts appear at least once in each of the unsorted libraries (Fig EV4, Appendix Table S1). When we select for library members that alter the NOT gate response functions, coverage of the library decreases to 50.2–74.8% in Bin 1, 38.5–50.0% in Bin 2, 25.0–26.1% in Bin 3, and 15.8–25.3% in Bin 4 (Fig EV4). This is expected since a limited subset of the constructs will be capable of shifting the gate thresholds. Indeed, > 50% of the composite parts encode promoters weaker than apFAB49, which generated less than a twofold change in RFP expression in our original experiments (Fig 1D).

### Sorted-parts strength analysis

Custom software ("IlluminaPerfSeqAnalysis.m", Code EV1) was written to analyze the perfect oligonucleotides, that is, sequences that are perfect matches to the designed library, sorted into each bin. Analysis relied on previously measured terminator (Chen *et al*, 2013) and promoter (Kosuri *et al*, 2013; Mutalik *et al*, 2013) strengths; therefore, all sequences with mutations were disregarded because they could change a part's activity and convolute the analysis. Occurrences of each promoter and terminator were counted per bin and used to calculate the median promoter, forward and reverse terminator strengths for each bin.

### Enrichment calculation

To calculate enrichment for each composite part, we normalized the counts of each composite part in a bin to the total number of perfect sequences in that bin. We defined the frequency of a composite part $f_{ijx}$ in a bin as

$$f_{ijx} = \frac{c_{ijx}}{\sum_i c_{ijx}} \tag{6}$$

where $c_{ij}$ is the number of occurrences of composite part $i$ in bin $j$ for library $x$, where $x = \text{PhlF}$, SrpR, or TarA. Then, we defined enrichment $E_{ijx}$ as the ratio of the frequencies of a composite part $i$ in a sorted bin $j$ to the frequency of that composite part in the unsorted pool ($f_{iux}$).

$$E_{ijx} = \frac{f_{ijx}}{f_{iux}} \tag{7}$$

If a composite part did not appear in the unsorted library at least once ($f_{iux} = 0$), $c_{iux}$ was set to one, indicating one count of the part in the unsorted library. This correction was used to ensure that none of the enrichments were infinite. Enrichment $E_{ijx}$ for each composite part was then averaged by bin across all three libraries. We defined the average enrichment $\overline{E_{ij}}$ for a composite part as

$$\overline{E_{ij}} = \frac{\sum_x E_{ijx}}{N} \tag{8}$$

where $N = 3$, the total number of libraries. To ensure that composite part behavior is consistent across all three libraries, any composite parts that did not appear in all three libraries for a given bin were assigned an enrichment of zero for that bin, that is, if $E_{ijx} = 0$ for any $x$, $\overline{E_{ij}} = 0$.

Next each composite part was assigned to the bin where its average enrichment was highest. Maximum average enrichment $E_{\max}$ for each composite part $i$ was calculated as

$$E_{\max_i} = \max_{1 \leq j \leq 4} (\bar{E}_{ij}) \tag{9}$$

Then, the composite part $i$ is assigned to the bin $j$, where $\overline{E_{ij}} = E_{\max i}$. If the maximum enrichment is less than one, the composite part is depleted in the sorted library and is not assigned to a bin (Fig 2F, gray). Depletion of a composite part in all of the sorted bins relative to the unsorted pool may be the result of biases in cell recovery after sorting or in amplification of the DNA for deep sequencing. The matrix of bin assignments was generated using custom software ("IlluminaEnrichmentGrid.m" Code EV1) and used to create Fig 2F.

### Measurement of growth curves

*Escherichia coli* strains were grown for 16 h in LB media containing antibiotics, when appropriate, in 96-deep well blocks (USA Scientific 1896–2000) at 37°C and 250 rpm in an INFORS-HT Multitron Pro. After 16 h, the cultures were diluted 1:200 into M9 medium with antibiotics and grown for 3 h with the same shaking and temperature settings as the overnight growth. Next, the cultures were diluted to $OD_{600} = 0.001$ into fresh M9 medium with antibiotics and 100 μM IPTG; 150 μl of these cultures was grown in black 96-well optical bottom plates (Thermo scientific 165305) at 37°C and 1 mm orbital shaking in a BioTek Synergy H1 plate reader. Optical density measurements at 600-nm wavelength ($OD_{600}$) were made every 20 min for 12 h.

## Construction and testing of the transcriptional interference model

Custom MATLAB software was written to solve the model ODEs (equations 3 and 4) with mixed boundary conditions. Initial mesh for the MATLAB boundary solver bvp4c was formed using MATLAB function bvpinit with general solutions for the model ODEs derived in Wolfram Alpha's Mathematica:

$$C_F(x) = \frac{C_1}{\varepsilon_R + C_2 e^{C_1 x}} \tag{10}$$

$$C_R(x) = \frac{C_1}{\varepsilon_F}\left(\frac{C_2 e^{C_1 x}}{\varepsilon_R + C_2 e^{C_1 x}}\right) \tag{11}$$

Integration constants were approximated using boundary conditions $C_F(x = 0) = \phi_F/v$ and $C_R(x = N) = \phi_R/v$. $\varepsilon_F$ and $\varepsilon_R$ were input directly into the model for $\varepsilon_F = \varepsilon_R = 1$ or parameter sweep experiments. Model results were reported as polymerase concentrations $C_F(x)$ and $C_R(x)$. Full-length transcript production is assumed to be proportional to $C_F|\varepsilon_F = 0$, which should be a measure of polymerases fired from $P_F$ that successfully transcribe the entire stretch of DNA between promoters. Fold repression $\theta$ is calculated using equation 5, which compares $C_F$ in the absence of interference ($C_F|\varepsilon_F = 0$) to $C_F$ with an interfering promoter ($C_F|\varepsilon_F > 0$). Model results for each forward/interfering promoter pair were scored by simple comparison to experimental data:

$$s = \left[\text{abs}\left(\theta_{TI\_m} - \theta_{TI\_p}\right)\right]^{-1} \tag{12}$$

where $s$ is the score for a specific promoter pairing, $\theta_{TI\_m}$ and $\theta_{TI\_p}$ are measured and predicted repression, respectively, for that pair. The best $\varepsilon_F$ and $\varepsilon_R$ values were calculated for each forward/interfering promoter pair using a weighted average, where each $\varepsilon$ was weighted by its score.

### Data availability

Brophy, JAN, Voigt, CA (2015). Antisense library. NCBI Sequence Read Archive SRP065456.

**Expanded View** for this article is available online.

## Acknowledgements

This work was supported by the Office of Naval Research (ONR) Multidisciplinary University Research Initiative (MURI grant 4500000552) and US National Science Foundation (NSF) Synthetic Biology Engineering Research Center (SynBERC EEC0540879). J.A.N.B is supported by an NSF Graduate Research Fellowship.

## Author contributions

CAV and JANB conceived of the study, designed the experiments, and wrote the manuscript. JANB performed the experiments and analyzed the data.

## Conflict of interest

The authors declare that they have no conflict of interest.

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
