## [Review Process File · Molecular Systems Biology]

Antisense transcription as a tool to tune gene expression

Jennifer A. N. Brophy

Corresponding author: Christopher A. Voigt, Massachusetts Institute of Technology

Review timeline:

Submission date:	31 August 2015
Editorial Decision:	28 September 2015
Revision received:	28 October 2015
Editorial Decision:	02 December 2015
Revision received:	04 December 2015
Accepted:	07 December 2015

Editor: Maria Polychronidou

Transaction Report:

1st Editorial Decision

28 September 2015

Thank you again for submitting your work to Molecular Systems Biology. We have now heard back from the three referees who agreed to evaluate your manuscript. Overall, the reviewers appreciate that the study presents a useful toolkit for synthetic biology as well as interesting biological insights. However, they raise a series of concerns, which should be carefully addressed in a revision of the manuscript. As you will see below, the referees' recommendations are rather clear.

If you feel you can satisfactorily deal with these points and those listed by the referees, you may wish to submit a revised version of your manuscript. Please attach a covering letter giving details of the way in which you have handled each of the points raised by the referees. A revised manuscript will be once again subject to review and you probably understand that we can give you no guarantee at this stage that the eventual outcome will be favorable.

REFeree COMMENTS

Reviewer #1:

The manuscript entitled "Antisense transcription as a means to tune the signal threshold required for gene expression." by Brophy and Voigt explore and exploit the processes surrounding antisense transcription and its effects on gene expression. They construct and characterize antisense promoters adjacent to transcriptional terminators, and show that they can reproducibly reduce protein expression in an anti-sense expression dependent manner. They then screen libraries of terminators and anti-sense promoter pairs to construct a catalogue of parts that could be used to tune existing expression in gene circuits. Finally, they do additional experiments and computational modeling to explore the mechanistic basis for the expression reduction.

Overall, the use of anti-sense promoters coupled to terminators could potentially be useful in tuning expression levels where engineering the promoter itself is challenging. I was impressed with the scope of the work as well including expression assays, library screenings, and mathematical modeling to develop both a set of well-behaved terminator-promoter pairings and a better mechanistic understanding for how the repression is occurring.

All that said, I feel there are two major issues that I have with the work that should be addressed. First, the Flow-seq and especially the analysis are not described in enough detail to adequately assess the work. From what I can tell, I am worried about several of the technical aspects of the experiment, but very little is described. For example, how many reads do you have per sample? How many reads pair, and how many are perfect in each bin? Why are so few of the constructs observed? Why aren't you correcting for the fact that you are sorting fewer cells in later bins? Why don't you use a weighted average across the bins as has been done in previous papers? Right now, it's pretty hard to assess the technical validity of this assay as it stands, and as pointed out in the minor comments, they should also do a better job of giving a statistical analysis of the data they are presenting. In addition, I will say that I worry about amplifying the library 30 cycles from plasmid DNA. It is important to at least state the amount of DNA going into the PCR reaction. Given the combinatorial nature, and structural homology, it's likely that there are significant errors building up during the amplification include PCR-template-switching issues. Finally, as is standard in the community, sequencing reads should be deposited in SRA, and all code should be released on a website such as GitHub. If code release is not possible, at the very least, a detailed description of what the algorithm is doing is necessary.

The second major comment is concerning the mechanistic modeling of what is happening here. I thought the trans anti-sense RNA expression experiment was very nice. However, I do worry about the interpretation of the results. The experiment indicates as you point out that the trans antisense transcription cannot account for the repression seen in cis. In fact, again as the authors point out it looks as though as the trans repression is very coincidentally the square root of the cis repression. You interpret this as saying this is because transcriptional interference and anti-sense RNA production have almost equal effect sizes, and assume that these are the only two possibilities. However, there are many other alternative explanations that seem more plausible. For instance, it could be that in cis, asRNA quickly forms double stranded RNA before ribosomes have a chance to bind; where-as in trans, by the time the RNA finds each other the sense RNA is already protected by translating ribosome. In such a scenario, the simplistic explanation would be that the fold repression in cis would only depend upon the asRNA concentration, while the trans repression would be dependent upon the product of the sense and antisense concentration, effectively giving a square root function. The larger point is that there are many explanations for the phenomena you are measuring (which was a great experiment to do), and the authors would benefit from a discussion of other hypotheses and perhaps future experiments that could distinguish from them. In general, I find the differential termination implausible from a mechanistic standpoint and the authors should discuss how such a scenario could occur (taking into account that the concept of momentum at low Reynold's number doesn't really apply).

Minor points:

Pg. 4. Is DH10B lacY-? If not, do you see any bimodal expression of GFP at intermediate to low lacY levels? Representative flow plots in the supplement would be helpful to see this.

Pg. 4. "Regulation by asRNA and transcriptional interference is not mutually exclusive" -> typo, 'is' should be 'are'

Pg. 5. Why do you add a terminator prior to antisense promoter here, when the above experiment was the opposite?

Pg 7. "There is a strong correlation between K and the fluorescence of the bin, but almost no effect on n." -> You need a statistical test here if you are going to make a statement like this.

Pg. 7. "Analysis of pooled constructs show that the ON and OFF states of all library members are essentially constant and threshold differences between the bins are statistically significant (Supplementary Figure 2)." -> Again, you need to give the name of the statistical test you used (and likely a p-value).

Pg. 7. There is no Figure 2g (I think this is a typo).

Pg. 7. "There is a strong correlation between the strength of the antisense promoter and the bin in which it appears (Figure 2e)." -> Again, statistical test.

- Pg. 8. "Composite parts that do not appear in all three libraries for a given bin were assigned an enrichment of zero for that bin. " -> This sentence is unclear; Are these the grey boxes in the figure? Does this mean that if the construct doesn't show up in any of the three experiments you zero it out?
- Pg. 8. Paragraph 2. When you use words like "enriched" or "correlate" please provide a statistical test
- Pg. 9. "parts tested are listed in Supplementary Information" -> I can't seem to find this.
- Pg. 9. "which is maintained at a copy number approximately two times higher" -> Do you account for this in plotting Figure 4a?
- Pg. 25. "The total fold-repression θ generated by the apFAB96 antisense promoter is" -> Why don't you show the other 3 promoters used (perhaps you could add them to the supplement)?

Reviewer #2:

The goal of this manuscript is to study systematically the effect of antisense transcription on the response function of a simple synthetic gene circuit. First, a terminator and an antisense promoter are inserted at the 5' UTR of an IPTG-inducible fluorescent reporter. The antisense promoter is found to repress the response function in general, and to increase its induction threshold. These effects increase with the strength of the antisense promoter. Next, a library of >5000 unique antisense promoter-terminator combinations are assembled and tested by flow-seq as parts of NOT gates utilizing 3 different repressors. This reveals features associated with efficient repression: antisense promoter strength, unidirectional termination, etc. Finally, a mathematical model reveals that the data can be matched best by a scenario where colliding polymerases do not always dissociate from the DNA.

Overall, this is a very interesting study introducing and characterizing a new toolkit for synthetic biology (technical advance) while revealing interesting biological (conceptual) knowledge about colliding polymerases. The manuscript should be published once the following comments are addressed.

Major points:

- (1) The flow-seq results mention that certain parts "do not appear in all three libraries for a given bin" and were assigned enrichment =0 for that bin. It would be useful to know more about these parts - their numbers, and some statistics about where they appear (in which bin) for various libraries. Are there any systematic trends? Can we learn something from this?
- (2) Collisions are modeled computationally based on mass-action kinetics (collisions proportional to the product of forward- and backward-elongating polymerases). The limitations of this assumption should be discussed, namely that two polymerases must collide and at least one of them must fall off before a different polymerase can collide. This suggests that some sort of queuing theory may be applicable.
- (3) The contribution from asRNA is probably still non-zero, and may be worth incorporating into the model to check its effects. It can be as simple as an RNA-RNA binding and sequestration process along with the other model components.
- (4) A discussion about the relevance and construction of similar eukaryotic systems would be useful. See also PMID:23385595.

Minor points:

- (5) The abstract should state what kind of organism (*Escherichia coli*) this study is based on.
- (6) The first word of the Introduction could be made more specific - maybe bioengineers?
- (7) The paper PMID: 20211838 investigated the effect of repressor expression on reporter response function in a yeast NOT gate. It may be relevant and may be worth citing.

(8) Fig. 2: Would the red line also indicate promoter activity at 100 uM IPTG in panel d? This should be clarified.

(9) There are several references to Figure 2, panel g, which does not exist.

(10) Check the grammar: "we employ here".

Reviewer #3:

The central idea of the manuscript is to use interference by antisense transcription as means to regulate the target gene expression in *E. coli*. This mode of gene regulation in prokaryotes has been relatively little studied and is not used in synthetic biology applications. The authors of this study performed a systematic high-throughput characterization of the effect of antisense transcription using a large library of antisense promoters of varying strengths in combination with unidirectional terminators. They demonstrated a strong correlation between the repression of the target gene and the strength of the antisense promoter. The authors described a panel of antisense promoters and unidirectional terminators producing the most predictable effect. In conclusion, they also constructed a quantitative model of interference by antisense transcription that fitted their data well.

The two main advances of the paper are: 1) systematic characterization of the set of promoter/terminator combinations that can be used in future synthetic circuits (technical advance); 2) quantitative model describing the effect of repression by antisense transcription given the size of the target gene and the promoter strength (conceptual advance).

The specific well-characterized genetic parts described in this paper will be of interest to synthetic biology community. The quantitative model of antisense interference will be useful for the biologists interested in the mechanism of transcription as well as to synthetic biologists looking to predict the effect of antisense transcription on their specific target genes.

Overall, the paper is interesting, contains potentially valuable practical insights for use in synthetic biology applications, and so could be published in MSB, provided that the major and minor points raised below are adequately addressed.

Major points:

- Effect of the promoters used for antisense repression on host cells. The constitutive antisense promoters used to regulate the target gene expression are all very strong. It would benefit the paper if authors valuated experimentally (preferably) or provided literature results on the effect of using such strong promoters as apFAB#96,78,140 on the cell growth. How likely is the load caused by these strong promoters to have other unintended effects on the host cell or synthetic circuits?
- Model assumptions: The assumptions used to derive the model of transcriptional interference should be clearly stated. What are specific requirements on the distributions of transcription initiation events in order for the model to be correct? For example, will the model be valid for promoters with strong bursting as opposed to the promoters with simple Poisson transcription initiation statistics. Ideally, the deterministic ODE model in eqs. (4) and (5) should be reduced from a more complex stochastic model that explicitly demonstrates the assumptions made. Alternatively, a direct stochastic simulation should be used to validate the deterministic model.
- Dependence of the repression on the length of the target gene: The model predicts a strong dependence of repression on the size of the target gene. For example for a weak target promoter the repression should scale as $\exp(\text{length of gene})$ according to the model (see below). It would greatly add to the value of the model if the authors validated this dependence experimentally, for example, by measuring fold repression upon addition of extra 5' UTR sequence to the target gene or by showing that in a bicistronic Ptac-GFP-RFP construct the RFP is stronger repressed by antisense transcription than GFP.

Minor points:

- Page 15. The authors relied on Wolfram Alpha to derive the solution of eqs. (4), (5). The expressions (7),(8) are formally correct, however they describe the real solutions of the model only

for imaginary values of the integration constant C_2 . The solution would look more natural as $CF(x) = C_1/(eR + C_2 * \exp(C_1*x))$, and the corresponding expression for $CR(x)$ obtained from the conservation law $eR*CF + eF*CR = C_1$. It is also not clear why they call (7),(8) "initial guesses" when in fact they are exact general solutions of eqs. (4),(5). We recommend the authors to correct the Methods section correspondingly.

- Page 9, paragraph 3. It is not clear why the effects of repression via asRNA and via antisense transcription interference are assumed to be multiplicative.
- Page 5, bottom paragraph. It is not clear why NOT-gate circuit was used in the experiments in Fig. 2, while a simpler reporter (Fig. 1A) could have probably been used to more directly measure the effect of various antisense promoter/terminator combinations on repression by antisense transcription.
- Page 7, paragraph 3. Why "...neither the forward nor antisense terminator strengths correlate strongly with changes in the response functions.."?

Presentation, style, and typos:

- Terminology. The terms like "promoter firing rate", "promoters fire", "RNAP is fired" used throughout the text are uncommon in literature and should be replaced or clarified. If the authors prefer to use this terminology, the meaning of the term "fire" must be clearly defined.
- Figure 1a. The cartoon inset should be made clearer. By convention, the top strand of the DNA is oriented 5' to 3' left to right. The direction of the DNA strands should also be indicated (5'-3').
- Figure 1a. Change RFP to rfp for consistency with the text and to adhere to E.coli gene naming convention.
- Figure 2a. Typo: "100 mM IPTG"
- Figure 4c,d. Describe the parameter q in the figure caption.
- Page. 4, paragraph 2. In the sentence containing "...in manner proportional.." change "proportional" to "monotonic" or another similar term.
- Page 4, paragraph 3. Change "After rfp..." to "Downstream of rfp..".
- Page 7, paragraph 4. Clearly define "gate thresholds".
- Page 8, paragraph 1. Change "Figure 2g" to "Figure 2f"
- Page 14, paragraph 2. Typo: "illumine" (presumably "Illumina").

1st Revision - authors' response

28 October 2015

Reviewer 1:

1. *First, the Flow-seq and especially the analysis are not described in enough detail to adequately assess the work. From what I can tell, I am worried about several of the technical aspects of the experiment, but very little is described. For example, how many reads do you have per sample? How many reads pair, and how many are perfect in each bin?*

The methods section has been substantially edited to include the requested information. This includes a Supplementary table (1) of statistics pertaining to the Illumina sequencing, a detailed description of the scripts used to analyze the sequences, and an Expanded View Figure (EV3) showing counts of composite parts in all three repressor libraries.

2. *Why are so few of the constructs observed?*

Approximately 70-77% of the composite parts are observed in the three unsorted libraries. There are several reasons why the remaining composite parts could be absent, including growth rate differences during library construction or circuit induction, sequence specific differences in oligonucleotide synthesis efficacy, and amplification bias during library construction or deep sequencing. The methods section was updated to discuss library coverage.

3. *Why aren't you correcting for the fact that you are sorting fewer cells in later bins? Why don't you use a weighted average across the bins as has been done in previous papers?*

Right now, it's pretty hard to assess the technical validity of this assay as it stands, and as pointed out in the minor comments, they should also do a better job of giving a statistical analysis of the data they are presenting. In addition, I will say that I worry about amplifying the library 30 cycles from plasmid DNA. It is important to at least state the amount of DNA going into the PCR reaction. Given the combinatorial nature, and structural homology, it's likely that there are significant errors building up during the amplification include PCR-template-switching issues.

We account for the fact that fewer constructs are sorted into the later bins by normalizing the frequency of a composite part in a bin to the total number of composite parts in that bin. We have edited the methods section to include a more detailed description of the data analysis and the amount of DNA going into the PCR reactions. We opted to assign composite parts to single bins instead of using a weighted average across bins since the simpler single assignment is enough to identify parts with differing abilities to alter the NOT gate response functions.

4. *Finally, as is standard in the community, sequencing reads should be deposited in SRA, and all code should be released on a website such as GitHub. If code release is not possible, at the very least, a detailed description of what the algorithm is doing is necessary.*

The sequencing reads and code were submitted to MSB along with the revised manuscript. As such, they will be publically accessible. In addition, we annotated the code so it should be easier to understand the algorithms and modify.

5. *The second major comment is concerning the mechanistic modeling of what is happening here. I thought the trans anti-sense RNA expression experiment was very nice. However, I do worry about the interpretation of the results. The experiment indicates as you point out that the trans antisense transcription cannot account for the repression seen in cis. In fact, again as the authors point out it looks as though as the trans repression is very coincidentally the square root of the cis repression. You interpret this as saying this is because transcriptional interference and anti-sense RNA production have almost equal effect sizes, and assume that these are the only two possibilities. However, there are many other alternative explanations that seem more plausible. For instance, it could be that in cis, asRNA quickly forms double stranded RNA before ribosomes have a chance to bind; where-as in trans, by the time the RNA finds each other the sense RNA is already protected by translating ribosome. In such a scenario, the simplistic explanation would be that the fold repression in cis would only depend upon the asRNA concentration, while the trans repression would be dependent upon the product of the sense and antisense concentration, effectively giving a square root function. The larger point is that there are many explanations for the phenomena you are measuring (which was a great experiment to do), and the authors would benefit from a discussion of other hypotheses and perhaps future experiments that could distinguish from them. In general, I find the differential termination implausible from a mechanistic standpoint and the authors should discuss how such a scenario could occur (taking into account that the concept of momentum at low Reynold's number doesn't really apply).*

When repression depends only upon the asRNA concentration, expression of the mRNA is completely silenced below a threshold of forward promoter activity. Differential expression of the asRNA changes only the threshold at which mRNA expression is observed and does not affect the repressed state of the system (Levine et al, 2007). Since the repressed state of the system changes with different antisense promoters in our experimental set up, repression in *cis* cannot be modeled as a first order reaction and an alternative mechanism must be considered to explain the repression in *cis*. Furthermore, the shape of the repression curves (Figure 1d) do not match repression patterns expected from asRNA alone. Systems regulated solely by asRNA become more sensitive to mRNA transcription rates at low levels of asRNA transcription, however we see that the weakest antisense promoter provides the most consistent repression across forward promoter activities (Figure 1d, blue). This behavior is consistent with previous models of transcriptional interference

(Sneppen et al, 2005). Thus, we believe that transcriptional interference contributes to the observed repression.

The modeling section of the paper has been edited to include other hypothesis and future experiments as well as a short discussion on the mechanisms that may underlie differential termination of RNAPs fired from convergent promoters.

6. *Pg. 4. Is DH10B lacY-? If not, do you see any bimodal expression of GFP at intermediate to low lacY levels? Representative flow plots in the supplement would be helpful to see this.*

DH10B is lacY-. We do not see bimodal expression of the fluorescent proteins. Representative cytometry data used to generate Figure 1c are now provided as Supplementary Figure 11.

7. *Pg. 4. "Regulation by asRNA and transcriptional interference is not mutually exclusive" -> typo, 'is' should be 'are'*

This has been corrected.

8. *Pg. 5. Why do you add a terminator prior to antisense promoter here, when the above experiment was the opposite?*

In the original system we quantify the impact of an antisense promoter on gene expression. In the second system, we expand on this design by adding the unidirectional terminator to demonstrate that antisense promoters can be added to outside of complete expression cassettes while still altering gene expression. The text has been edited to clarify this point.

9. *Pg 7. "There is a strong correlation between K and the fluorescence of the bin, but almost no effect on n. " -> You need a statistical test here if you are going to make a statement like this.*

Two sample student t-tests were performed to assess differences in K and n between bins. The results of these tests are now available as Expanded View Figure 1 (EV1).

10. *Pg. 7. "Analysis of pooled constructs show that the ON and OFF states of all library members are essentially constant and threshold differences between the bins are statistically significant (Supplementary Figure 2). " -> Again, you need to give the name of the statistical test you used (and likely a p-value).*

One-way ANOVA tests with Tukey HSD post hoc tests were performed on one-hundred randomly selected data points from each sample. The results of these tests have been added as Expanded View Figure 2 (EV2).

11. *Pg. 7. There is no Figure 2g (I think this is a typo).*

This was indeed a typo and has been corrected.

12. *Pg. 7. "There is a strong correlation between the strength of the antisense promoter and the bin in which it appears (Figure 2e)." -> Again, statistical test.*

Graphs of bin fluorescence vs. part strengths were added as Supplementary Figure 3 with R^2 values.

13. *Pg. 8. "Composite parts that do not appear in all three libraries for a given bin were assigned an enrichment of zero for that bin. " -> This sentence is unclear; Are these the grey boxes in the figure? Does this mean that if the construct doesn't show up in any of the three experiments you zero it out?*

The grey boxes in Figure 2f represent composite parts that are not assigned to a bin. Composite parts are not assigned to bins when their maximum average enrichment is less than one, indicating the composite part is not represented in at least one bin across all three libraries part or that it is depleted in the all of the sorted bins relative to the unsorted library. Depletion of a composite part in all of the sorted bins relative to the unsorted pool may be the result of biases in cell recovery after sorting or differences in amplification of the DNA for deep sequencing. Thus these parts were omitted from our analysis. The text and figure caption have been edited to clarify this point. In addition, a detailed description of the enrichment calculations was added to the methods section to clarify the library analysis.

14. Pg. 8. Paragraph 2. When you use words like "enriched" or "correlate" please provide a statistical test

This paragraph is a qualitative description of the heat map in Figure 2f. The language has been edited to reflect this.

15. Pg. 9. "parts tested are listed in Supplementary Information" -> I can't seem to find this.

The parts are listed in Supplementary Table 2. The manuscript has now been edited to say "listed in Supplementary Table 2".

16. Pg. 9. "which is maintained at a copy number approximately two times higher" -> Do you account for this in plotting Figure 4a?

No, we do not make adjustments for the copy number. We draw attention to the difference in copy number so it is clear that the asRNA is being over expressed in *trans* relative to its abundance in *cis*.

17. Pg. 25. "The total fold-repression θ generated by the *apFAB96* antisense promoter is" -> Why don't you show the other 3 promoters used (perhaps you could add them to the supplement)?

Data on the other 3 promoters is now provided as Supplementary Figure 12.

Reviewer 2:

1. The flow-seq results mention that certain parts "do not appear in all three libraries for a given bin" and were assigned enrichment =0 for that bin." It would be useful to know more about these parts - their numbers, and some statistics about where they appear (in which bin) for various libraries. Are there any systematic trends? Can we learn something from this?

An Expanded View Figure (EV3) was added to the manuscript to show the abundance of every composite part in the repressor libraries by bin. A paragraph was added to the methods section to discuss library coverage.

2. Collisions are modeled computationally based on mass-action kinetics (collisions proportional to the product of forward- and backward-elongating polymerases). The limitations of this assumption should be discussed, namely that two polymerases must collide and at least one of them must fall off before a different polymerase can collide. This suggests that some sort of queuing theory may be applicable.

A more detailed discussion of the model assumptions was added to the results and discussion sections.

3. *The contribution from asRNA is probably still non-zero, and may be worth incorporating into the model to check its effects. It can be as simple as an RNA-RNA binding and sequestration process along with the other model components.*

We agree that the contribution from asRNA may be non-zero and address this by emphasizing the assumptions made to fit our model to the experimental data in the text.

4. *A discussion about the relevance and construction of similar eukaryotic systems would be useful. See also PMID:23385595.*

A short discussion on applicability to eukaryotic systems was incorporated into the discussion along with references.

5. *The abstract should state what kind of organism (Escherichia coli) this study is based on.*

This has been edited as suggested.

6. *The first word of the Introduction could be made more specific - maybe bioengineers?*

This has been edited to "Genetic engineers."

7. *The paper PMID: 20211838 investigated the effect of repressor expression on reporter response function in a yeast NOT gate. It may be relevant and may be worth citing.*

This was incorporated as suggested.

8. *Fig. 2: Would the red line also indicate promoter activity at 100 uM IPTG in panel d? This should be clarified.*

Yes, the red line represents Ptac activity at 100uM IPTG, which is the same in panels b and d. The line was omitted in panel d for visual clarity.

9. *There are several references to Figure 2, panel g, which does not exist.*

This was a typo that should have read "Figure 2f" and it has been corrected.

10. *Check the grammar: "we employee here".*

This typo has been corrected.

Reviewer 3:

1. *Effect of the promoters used for antisense repression on host cells. The constitutive antisense promoters used to regulate the target gene expression are all very strong. It would benefit the paper if authors valuated experimentally (preferably) or provided literature results on the effect of using such strong promoters as apFAB#96,78,140 on the cell growth. How likely is the load caused by these strong promoters to have other unintended effects on the host cell or synthetic circuits?*

Cell growth experiments are now provided as Supplementary Figure 13.

2. *Model assumptions: The assumptions used to derive the model of transcriptional interference should be clearly stated. What are specific requirements on the distributions of transcription initiation events in order for the model to be correct? For example, will the model be valid for promoters with strong bursting as opposed to the promoters with simple Poisson transcription initiation statistics. Ideally, the deterministic ODE model in eqs. (4) and (5) should be reduced from a more complex stochastic model that explicitly*

demonstrates the assumptions made. Alternatively, a direct stochastic simulation should be used to validate the deterministic model.

The model assumes that transcription initiation rates are Poisson distributed and is not able to account for differences in gene expression that may occur because of promoters with strong bursting. The parameter e may reflect either the ability of polymerases to survive collision or the avoidance of collision altogether to due promoter bursting. This point has been added to the model presentation along with a reference to a more complex stochastic model of transcriptional interference, which can be simplified to the model presented here.

- 3. Dependence of the repression on the length of the target gene: The model predicts a strong dependence of repression on the size of the target gene. For example for a weak target promoter the repression should scale as $\exp(\text{length of gene})$ according to the model (see below). It would greatly add to the value of the model if the authors validated this dependence experimentally, for example, by measuring fold repression upon addition of extra 5' UTR sequence to the target gene or by showing that in a bicistronic Ptac-GFP-RFP construct the RFP is stronger repressed by antisense transcription than GFP.*

Additional experiments have been added to test the dependence of repression on length of the DNA between the forward and antisense promoter. Indeed, the expected behavior is observed. This data has been added to Figure EV4.

- 4. Page 15. The authors relied on Wolfram Alpha to derive the solution of eqs. (4), (5). The expressions (7),(8) are formally correct, however they describe the real solutions of the model only for imaginary values of the integration constant C2. The solution would look more natural as $CF(x) = C1/(eR + C2 * \exp(C1*x))$, and the corresponding expression for $CR(x)$ obtained from the conservation law $eR*CF + eF*CR = C1$. It is also not clear why they call (7),(8) "initial guesses" when in fact they are exact general solutions of eqs. (4),(5). We recommend the authors to correct the Methods section correspondingly.*

We have updated the methods section.

- 5. Page 9, paragraph 3. It is not clear why the effects of repression via asRNA and via antisense transcription interference are assumed to be multiplicative.*

At steady-state, the concentration of mRNA will be its production rate divided by the degradation rate. Transcriptional interference affects the production rate and asRNA affects its degradation. While mechanisms could be imagined that couple these effects, the simplest expectation is that they would be multiplicative.

- 6. Page 5, bottom paragraph. It is not clear why NOT-gate circuit was used in the experiments in Fig. 2, while a simpler reporter (Fig. 1A) could have probably been used to more directly measure the effect of various antisense promoter/terminator combinations on repression by antisense transcription.*

The NOT-gate circuit was used instead of the simpler reporter to demonstrate that antisense promoters can be used to tune more complex genetic circuits. One of the goals of this work is to understand the potential role of antisense transcription on regulation and this gate is one of the simplest model genetic circuits.

- 7. Page 7, paragraph 3. Why ".neither the forward nor antisense terminator strengths correlate strongly with changes in the response functions.."?*

There are stronger features of the terminators that correlate with changes in the response functions. While forward and reverse terminator strength may influence how much antisense transcription of a gene can occur, the correlation between response function changes and terminator identity is dominated by the presence of other terminator features (such as cryptic antisense promoter activity or mRNA destabilization), therefore there is no change in the average terminator strengths across sorted bins.

8. *Terminology. The terms like "promoter firing rate", "promoters fire", "RNAP is fired" used throughout the text are uncommon in literature and should be replaced or clarified. If the authors prefer to use this terminology, the meaning of the term "fire" must be clearly defined.*

This has been edited as suggested.

9. *Figure 1a. The cartoon inset should be made clearer. By convention, the top strand of the DNA is oriented 5' to 3' left to right. The direction of the DNA strands should also be indicated (5'-3').*

The figure has been edited as suggested.

10. *Figure 1a. Change RFP to rfp for consistency with the text and to adhere to E.coli gene naming convention.*

This has been edited as suggested.

11. *Figure 2a. Typo: "100 mM IPTG"*

We have ensured that this is corrected.

12. *Figure 4c,d. Describe the parameter q in the figure caption.*

This has been edited as suggested.

13. *Page 4, paragraph 2. In the sentence containing "...in manner proportional.." change "proportional" to "monotonic" or another similar term.*

This has been edited as suggested.

14. *Page 4, paragraph 3. Change "After rfp..." to "Downstream of rfp..".*

This has been edited as suggested.

15. *Page 7, paragraph 4. Clearly define "gate thresholds".*

This has been edited as suggested.

16. *Page 8, paragraph 1. Change "Figure 2g" to "Figure 2f"*

This typo has been corrected.

17. *Page 14, paragraph 2. Typo: "illumine" (presumably "Illumina").*

This typo has been corrected.

Thank you again for submitting your work to Molecular Systems Biology. We have now heard back from the two referees who were asked to evaluate your manuscript. As you will see below, the referees think that their major concerns have been satisfactorily addressed. However, referee #1 still lists two concerns, which we would ask you to address in a revision of this work. In particular, we would ask you to make sure that the statistical support for the presented findings and all information on the related statistical analyses (i.e. statistical test, the actual P value for each test etc.) is provided

in the main text and the figure legends. Moreover, we would welcome the inclusion of a short discussion on potential alternative hypotheses (point #2 of reviewer #1).

REFeree COMMENTS

Reviewer #1:

As in the first review, overall I'm impressed with the work and believe it should be published in MSB. I especially am happy that the authors are including the code and datasets to be able to better aid reproducibility. I still think there are issues that could be improved both on the statistics and modeling fronts. I will leave it to the authors and editors to decide how much to improve the final manuscript, but here are my most salient issues.

1. Significance tests - There are still many areas of the text that claim correlations or differences without any statistical test. I think the text would be helped a lot by just adding the significance (or goodness of fit) in parentheses after the sentence rather than having to refer to the supplement or paper. Beyond that, it is very confusing to have different levels of significance and insignificance displayed as increasing number of stars. It would be much better to either display the p value itself, or use a more standard annotation (e.g., n.s. for not significant). I would also get rid of the $p < 0.10$ indication and label it n.s.

2. Modeling - I still think that the modeling section does not take into account other possibilities including cis versus trans repression for the anti-sense transcripts. I think the authors would still benefit from an increased discussion about the model's limitations, especially because other hypotheses beyond those that are presented are not considered. For example, I agree with the authors that as their model is structured, that both asRNA and TI have to be invoked. However, there are other possible mechanisms or model structures that can explain the effects. For example, if asRNA acts differentially in cis and trans could that explain the differential effects seen; or that asRNA acts only at the ribosome binding site, or any number of other scenarios.

Reviewer #3:

We found the revised manuscript and the author's responses to the critiques adequate, and recommend publication.

2nd Revision - authors' response

04 December 2015

We are resubmitting our paper "Antisense transcription as a tool to tune gene expression" for consideration for publication in Molecular Systems Biology.

We have made the following revisions as requested:

- Statistical tests, p-values and R^2 values for all findings have been added to the main text and figure legends.
- A short discussion on alternative mechanisms for *cis* repression of gene expression was included in the discussion (main text pg. 13, paragraph 1).